# Neuronal tuning to threat exposure remains stable in the mouse prefrontal cortex over multiple days

Ole Christian Sylte[1,2], Hannah Muysers[1,2], Hung-Ling Chen[1], Marlene Bartos[1], Jonas-Frederic Sauer[1]*

1 University of Freiburg, Medical Faculty, Institute of Physiology I, Freiburg, Germany, 2 University of Freiburg, Faculty of Biology, Freiburg, Germany

* jonas.sauer@physiologie.uni-freiburg.de

## Abstract

Intense threat elicits action in the form of active and passive coping. The medial prefrontal cortex (mPFC) executes top-level control over the selection of threat coping strategies, but the dynamics of mPFC activity upon continuing threat encounters remain unexplored. Here, we used 1-photon calcium imaging in mice to probe the activity of prefrontal pyramidal cells during repeated exposure to intense threat in a tail suspension (TS) paradigm. A subset of prefrontal neurons displayed selective activation during TS, which was stably maintained over days. During threat, neurons showed specific tuning to active or passive coping. These responses were unrelated to general motion tuning and persisted over days. Moreover, the neural manifold traversed by low-dimensional population activity remained stable over subsequent days of TS exposure and was preserved across individuals. These data thus reveal a specific, temporally, and interindividually conserved repertoire of prefrontal tuning to behavioral responses under threat.

## Introduction

Prefrontal pyramidal neurons display pronounced complexity in their activities. For instance, many cells respond in a nonlinear fashion to combinations of task variables, a property termed "mixed selectivity" [1,2]. Moreover, prefrontal activity is highly dynamic. This is apparent in tasks in which animals learn new rules: When a previously learned rule is reinstated after a rule change, the prefrontal population settles to a novel activity state [3]. These results suggest that the medial prefrontal cortex (mPFC) network provides an abstract and dynamic representation of task structure and rules rather than a rigid set of response commands. Besides the extensively studied involvement in cognitive tasks, the mPFC executes prominent top-level control over stereotyped, learned behaviors such as the acquisition [4,5] and extinction [6,7] of fear responses and acquired threat evasion [8], and over innate responses such as dominance behavior [9] and escape from imminent threats [10], which are key to animal survival. While theoretical work has emphasized the advantages of complex and dynamic coding regimes for

**Data Availability Statement:** The dataset generated during the current study, source data, and analysis code are available at https://doi.org/10.5281/zenodo.10378756.

**Funding:** This work was supported by Else Kröner-Fresenius Stiftung (https://www.ekfs.de/, grant 2019_A173 to J.-F.S.), German Research Foundation (Deutsche Forschungsgemeinschaft, https://www.dfg.de/, grant SA3609/1-1 to J.-F.S.), German Research Foundation (Deutsche Forschungsgemeinschaft, https://www.dfg.de/, FOR5159 - TP7 to J.-F.S (grant SA3609/2-1 to J.-F.S. and grant BA1582/16-1 to M.B.). The funders had no role in study design, data collection and analysis, decision to publish, or preparation of the manuscript.

**Competing interests:** The authors have declared that no competing interests exist.

**Abbreviations:** DAPI, 4′-6-diamidino-2-phenylindole; EV, explained variance; FS, forced swimming; GRIN, gradient reflective index; mPFC, medial prefrontal cortex; PBS, phosphate-buffered saline; PETH, peri-event time histogram; PFA, paraformaldehyde; PV, parvalbumin; SOM, somatostatin; SVD, singular value decomposition; TS, tail suspension.

cognitive flexibility [1,11], it is far less intuitive how simple, stereotypical behavioral responses might be controlled by dynamic prefrontal population activity.

One such innate behavior under the control of the mPFC is the response to intense threat in a tail suspension (TS) paradigm. In this task, mice are elevated by their tail for a fixed amount of time and are thus forced in an inescapable, stressful situation. During TS, mice display periods of active struggle (i.e., intense movement) interspersed with passive coping (i.e., immobility) [12]. Prefrontal networks have been implicated in controlling behavioral responses during such threatful states. First, the firing of a subset of prefrontal units selectively signals movement in the TS but not general movement in a neutral open field [13]. Second, deep brain stimulation of mPFC in rats reduces active escape responses during forced swimming (FS), a paradigm closely related to TS [14]. Third, TS induces expression of the immediate early gene cFos in the mPFC, suggesting the activation of prefrontal neurons during TS [15]. Fourth, optogenetic silencing of mPFC neurons projecting to dorsal raphe nucleus and lateral habenula reduce and enhance, respectively, active coping responses of rats during FS [10], and closed-loop stimulation of synaptically connected PFC-thalamic loops reduces TS struggling in mice [13]. Finally, functional impairments of the mPFC on the synaptic, cellular, and network levels have been observed in mouse models as well as in patients who have psychiatric illness with impaired coping responses [16–20].

Despite the established role of the mPFC in controlling coping responses, few studies have directly investigated mPFC activity during TS [13,21–23] or related paradigms [10]. Crucially, those studies relied on electrophysiological recordings [13,22,23], thus limiting the analysis to "snapshots" of the activity of identified neurons on single recording days, or utilized calcium imaging from defined interneuron types [21] during TS exposure. As a result, the dynamics with which threat-related activity of prefrontal pyramidal neurons evolves on the timescale of days have so far not been investigated. Here, we utilized calcium imaging during repeated TS exposures on multiple days to overcome this limitation. We identify a rigid coding regime in which the activity of individual neurons and the dynamics of low-dimensional population responses are highly conserved over multiple TS exposures and across individuals.

## Results

### Longitudinal calcium imaging during high threat in the TS

To dissect the dynamics of prefrontal neuronal responses during intense threat, we performed 1-photon $Ca^{2+}$ imaging from prefrontal (prelimbic and cingulate) layer 5 pyramidal cells in transgenic Thy1-GCaMP6f mice during exposure to daily sequences of homecage (baseline) and TS (performed on days 1, 2, 3, and 9 with imaging on days 1, 3, and 9, $n = 6$ mice; **Figs 1A, 1B, and S1**). Under TS exposure, mice displayed periods of struggling and immobility (**Fig 1C**), which allowed us to assess neuronal activities in relation to both active and passive coping styles. Consistent with previous reports [13], the time spent passively coping increased over the first TS exposures (**Fig 1C**). On each recording day, comparable numbers of cells were detected ($n = 1,382$, 1,367, and 1,518 on days 1, 3, and 9, respectively, from $n = 6$ mice). When registering neurons over days, we found that 39.6% (SD = 4.95) of cells were detected as active on all 3 imaging days (we call this population "repeatedly identified neurons"; **Figs 1D and S2**). Repeatedly identified neurons showed highly correlated signal-to-noise ratios over successive imaging days (**Fig 1D**). These results suggest that 1-photon imaging enables stable recordings from the same set of prefrontal neurons over multiple TS exposures.

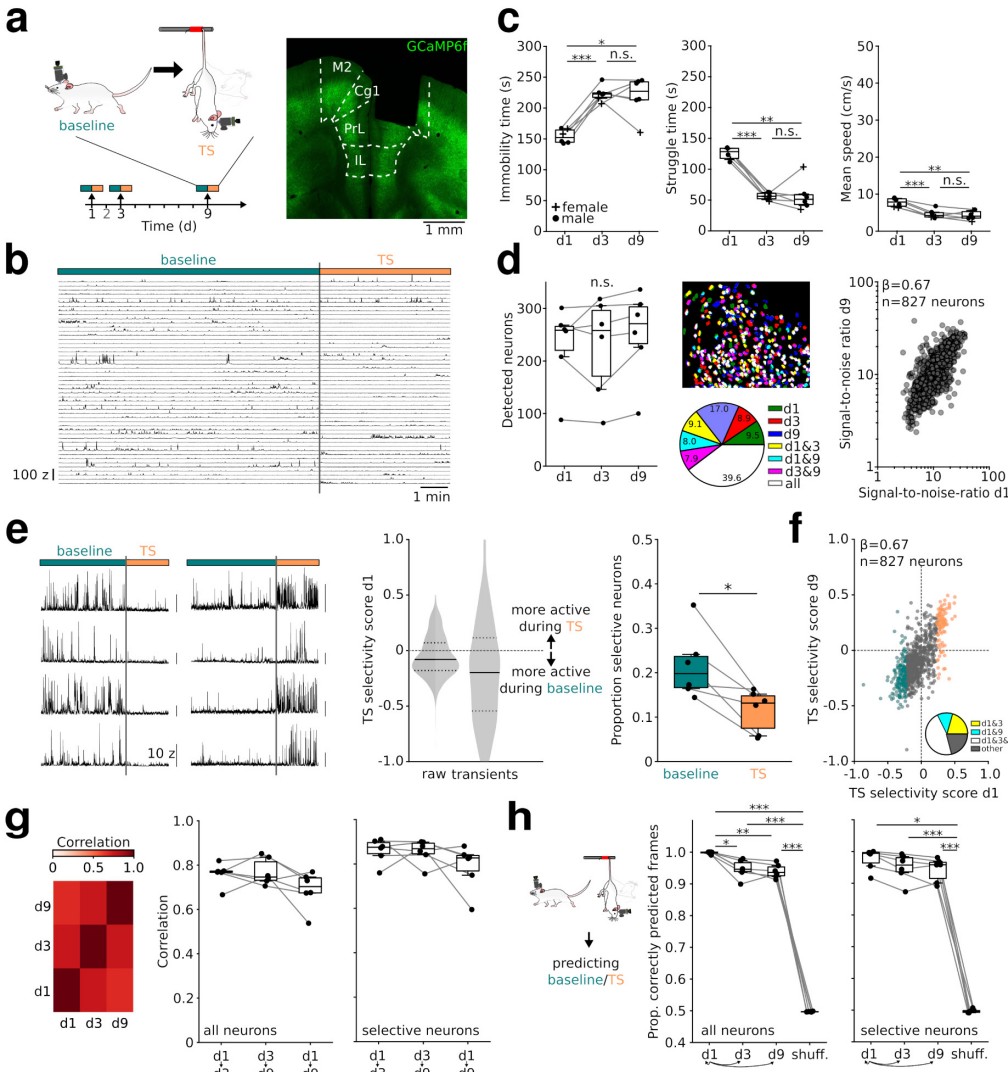

**Fig 1. Stable activity of prefrontal neurons upon repeated TS exposure. (a)** Schematic of 1-photon recording from mPFC pyramidal neurons during TS. A sequence of baseline and TS was recorded on 3 days (d1, d3, and d9). Right: Example of lens location in the mPFC. M2: secondary motor cortex, Cg1: cingulate cortex, PrL: prelimbic cortex, IL: infralimbic cortex. GCaMP6f (green) is predominantly expressed in deep layer pyramidal cells. **(b)** Z-scored calcium activity of 50 randomly selected neurons of one mouse during baseline and TS on d1. **(c)** Time-dependent changes in TS behavior. Compared to d1, the animals spent more time during passive coping on d3 (immobility time: t = 10.93, p = 0.0003, struggle time: t = 15.81, p = 5 * 10^{-5}, mean speed: t = 9.8, p = 0.0006) and d9 (immobility time: t = 4.32, p = 0.023, struggle time: t = 5.4, p = 0.009, mean speed: t = 5.65, p = 0.007). The coping responses did not differ between d3 and d9 (immobility time: t = 0.31, p = 1, struggle time: t = 0.1, p = 1, mean speed: t = 0.79, p = 1). **(d)** Left: Comparable numbers of neurons are detected per mouse across imaging days (F = 1.87, p = 0.2). Middle: example field of view from multisession alignment with cells active on single or multiple days color-coded. Right: signal-to-noise ratio of neurons from all mice found active on all days (termed "repeatedly identified neurons") remain correlated on d1 and d9, β = 0.67 ± 0.06, p < .001, linear mixed effects model (LME). **(e)** Left: Examples of baseline- and TS-selective neurons. Middle: TS selectivity score computed with raw signals and transient rates (−1: active only during baseline; 1: active only during TS) on d1 for all 1,382 neurons. Solid lines show median, dashed lines the first and third quartiles. Right: proportion of neurons with TS selectivity score <−0.2 ("baseline selective") and >0.2 ("TS selective," based on raw signals). More neurons were found to be baseline selective. t = 3.22, p = 0.02, paired *t* test, *n* = 6 mice. **(f)** Correlation of TS selectivity score of all repeatedly identified neurons on d1 and d9. β = 0.67 ± 0.09, p < .001, LME. Color code as in (**e**). Inset: Proportion of neurons scored as selective on day 1 that also remained selective on days 3, 9, or both. **(g)** Summary of correlation of TS selectivity scores across days (mouse averages) for all neurons (F = 3.22, p = 0.083) and for selective neurons only (F = 2.33, p = 0.148). **(h)** Prediction of baseline and TS state on a frame-by-frame basis with models trained exclusively on calcium data of d1. Accuracy across days was lower compared to within-d1 (d3: t = 4.61, p = 0.029, d9: t = 7, p = 0.005) but above chance level (vs. shuffled: d1: t = 188.76, p = 10^{-9}, d3: t = 36.27,

$p = 10^{-6}$, d9: $t = 51.69$, $p = 10^{-7}$). Chance level was determined by running the analysis with randomly time-shifted calcium traces. Similar results were obtained for models trained on selective neurons only (d1 vs. d3: $z = 1.6$, $p = 0.547$, d1 vs. d9: $z = 1.6$, $p = 0.547$, d1 vs. shuffle: $z = 2.88$, $p = 0.02$, d3 vs. shuffle: $t = 23.14$, $p = 10^{-5}$, d9 vs. shuffle: $t = 27.44$, $p = 6 * 10^{-6}$). (c, d, g, h): One-way repeated measures ANOVAs followed by paired $t$ tests or Wilcoxon rank sum tests with Bonferroni correction, $n = 6$ mice. *$p < 0.05$, **$p < 0.01$, ***$p < 0.001$. The data underlying this figure can be found at https://doi.org/10.5281/zenodo.10378756.

## A subset of prefrontal neurons stably activates during intense threat over successive days

To quantify the responses of prefrontal neurons to TS threat, we computed a "TS selectivity score" based on calcium signals for each neuron. This metric is bound to vary from −1 (activity only detected during baseline) to 1 (activity only found during TS). The distribution of TS selectivity scores was skewed, with many neurons showing preferential activity during baseline (277 cells with score $<-0.2$, approximately 20% of the active population) and fewer cells being selective for TS (160 cells with score $>0.2$, approximately 12% of the active population; Fig 1E). Similar results were obtained when we computed TS selectivity based on significant transients (Fig 1E) and when the data were separately analyzed for neurons located in prelimbic and cingulate areas or for neurons from female and male mice (S3A and S3C Fig).

We next asked whether prefrontal neurons retain their individual TS selectivity over subsequent days. Considering repeatedly identified neurons, we computed the correlation between TS selectivity scores across days. This analysis revealed highly stable TS selectivity of the population (Fig 1F and 1G). Moreover, a large fraction of neurons with baseline or TS selectivity on day 1 remained selective on days 3 and 9 (Figs 1F, S3B and S3D). Consistently, logistic regression models trained on calcium data of the first day could significantly decode the baseline/TS state on a frame-by-frame basis with calcium data of subsequent days (Fig 1H). While decoding was most accurate when train and test data were taken from within day 1, accuracy of testing on data from days 3 and 9 remained high ($>0.9$; Fig 1H). Behavioral state decoding over days was robust against the choice of the decoding model (S3E Fig). These data jointly suggest that prefrontal neurons retain a stable preference for baseline and TS over time.

## Representation of active and passive coping during threat

We next investigated whether the activity of prefrontal neurons contains information about active and passive coping responses during threat. Before turning to the dynamics of prefrontal responses over repeated TS exposure, we initially focused this analysis on the first day of TS to obtain a basic understanding of coping-related neuronal responses. We classified the animals' behavior during TS into binary states of immobility (i.e., passive coping) and struggling (i.e., active coping; Fig 2A and S1 Video). To test whether the current active/passive behavioral state could be predicted from calcium data on a frame-by-frame basis, we trained a logistic regression model using a balanced cross-validation regime. This analysis revealed high accuracy, significantly exceeding predictions based on time-shifted control data (Fig 2A). Similar results were obtained using various decoding models (S4A–S4C Fig). The ability to decode the current behavioral state was based on the distinct preference of individual neurons toward struggling or passive states (Fig 2B), consistent with previous electrophysiological studies on stress coping responses [10,13]. This was quantified with a "movement score" defined as the Pearson's correlation coefficient between the animal's speed during TS and the calcium trace of each neuron. We found a broad distribution of movement scores for TS, with many neurons showing large positive or negative scores, corresponding to a strong tuning to motion and immobility, respectively. In contrast, during baseline conditions, movement score

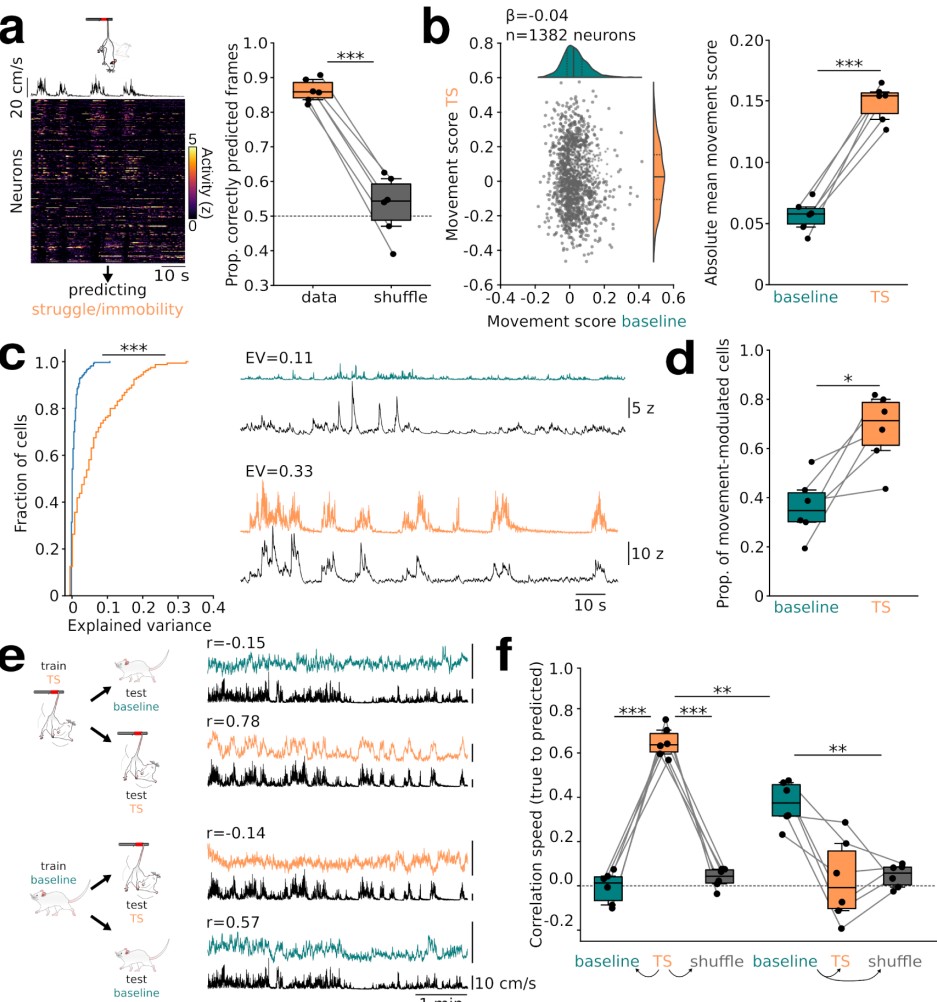

**Fig 2. Rate coding of passive and active coping responses. (a)** Decoding of TS struggle/immobility from calcium data. Left: Simultaneously recorded neurons ($n$ = 301) in one mouse during the first day of TS exposure (sorted by their correlation to TS movement) along with movement speed (top). Right: Results of decoding with real vs. surrogate data created by randomly shifting calcium traces in time (t = 11.26, p = 9 * 10$^{-5}$, paired $t$ test, $n$ = 6 mice). **(b)** Correlation of d1 movement scores during TS and baseline (i.e., standardized β-coefficient between the animal's speed and each neuron's calcium trace). Distributions of baseline and TS movement scores are shown on the top and side, respectively. Solid lines show the median, dashed lines the first and third quartiles. β = −0.04 ± 0.05, p = 0.423, LME. Right: Stronger movement tuning during TS as quantified from the average absolute movement score of each mouse. t = −15.0, p = 10$^{-5}$, paired $t$ test, $n$ = 6 mice. **(c)** Linear model predicting calcium signals of individual neurons with movement. Left: Summary of explained variance (EV) during baseline ($n$ = 277 cells) and TS ($n$ = 160 cells, U = 9,313, p = 10$^{-24}$, Mann–Whitney U test). Right: Examples of true (black) and predicted calcium signals (colored) during both states. **(d)** Proportion of neurons for which calcium activity could be significantly explained by movement alone (t = −3.82, p = 0.012, paired $t$ test). **(e)** Decoding baseline and TS motion speed with models trained on TS (top) or baseline (bottom) population calcium data. Examples of one mouse are shown with the true speed in black and the predicted speed in color. Pearson's correlation coefficient for each fit is shown on top. **(f)** Quantification of the data shown in **(e)** for all mice. Trained on TS: TS vs. baseline: t = 12.43, p = 0.0004, TS vs. shuffle: t = 13.77, p = 0.0003, baseline vs. shuffle: t = 2.51, p = 0.375. Trained on baseline: baseline vs: TS: t = 3.89, p = 0.08, baseline vs. shuffle: t = 8.28, p = 0.003, TS vs. shuffle: t = 0.26, p = 1.0. Trained on TS predicting TS vs. trained on baseline predicting baseline; t = 8.09, p = 0.0033, one-way repeated measures ANOVA followed by paired $t$ tests with Bonferroni correction. Dashed lines: chance level. $n$ = 6 mice, *p < 0.05, **p < 0.01, ***p < 0.001. The data underlying this figure can be found at https://doi.org/10.5281/zenodo.10378756.

distributions were narrower, with fewer neurons taking extreme values, as evidenced from a larger rectified movement score during TS (**Fig 2B**). Importantly, movement scores during TS were uncorrelated to the same neurons' scores during baseline, suggesting that activity during active/passive coping is not simply a reflection of general motion tuning (**Fig 2B**). Two hypotheses emerge from these observations: that the activity of prefrontal neurons carries more information about the current movement state during TS than during baseline and that this information does not generalize between baseline and TS movement. We further tested these hypotheses as follows: First, we trained a linear regression model to predict calcium signals of individual baseline- and TS-active neurons from movement traces. Movement explained more of the variance in calcium signal during TS (6.23 ± 0.05%) than baseline (0.74 ± 0.08%; **Fig 2C**). Consequently, the fraction of neurons for which variations in their calcium signal could be significantly explained by ongoing movement was larger during TS (**Fig 2D**). In line with the stronger and more extreme neuronal correlations to ongoing movement, a linear model taking the activity of other cells as variables explained more of the variance of individual neuron's calcium signal during TS compared to baseline (**S4D–S4F Fig**). Consistent with previous electrophysiological data [13], a fraction of cells already elevated their activity before behavioral transitions from struggling to immobility and vice versa (**S4G and S4H Fig**). Second, we trained separate decoders on population calcium data from either baseline or TS and predicted movement both within (i.e., baseline → baseline and TS → TS) and across conditions (i.e., baseline → TS and TS → baseline; **Fig 2E**, see **S4I Fig** for predictions of movement of different body parts). Within-condition, both baseline and TS decoders performed significantly against surrogate data created by randomly shifting each neuron's activity in time (**Fig 2E and 2F**). However, the decoder was significantly more effective at predicting TS compared to baseline speed (**Fig 2F**). In contrast, for both baseline-TS and TS-baseline cross-decoding conditions, the models performed at chance level (**Fig 2F**). A TS coding regime that differs from general motion tuning during baseline thus discriminates between active and passive coping styles.

## Neuronal tuning to threat-related movement is stable over time

Repeated exposure to intense threat over subsequent days consistently elicits stereotyped active and passive coping responses, raising the question whether the same neurons maintain their individual tuning to threat-related movement over time. Utilizing repeatedly identified neurons ($n = 827$), we assessed the proportion of struggling- and immobility-tuned neurons (defined by a TS movement score of $>0.2$ and $<-0.2$, respectively). These proportions remained similar over days, albeit a small increase in the proportion of struggle-selective neurons was observed (**S5A Fig**). The mean activity of struggle-selective cells during struggling increased over days while that of immobility-selective neurons during immobility decreased (**S5B Fig**). Despite these changes in mean activity, we found significant correlations in the neurons' movement score over successive TS days, suggesting that neurons maintain stable tuning to coping-related movement (**Fig 3A and 3B**). Stable tuning was apparent within a single recording day, quantified as the correlation in movement score between the first and second half of TS exposure (**S5C Fig**). In addition, stable tuning over days did depend neither on the animal's sex nor on the anatomical location in the prelimbic or cingulate cortex (**S5D and S5E Fig**). Behavioral state within TS (i.e., struggle versus immobility) as well as changes in the animals' speed during TS on days 3 and 9 could be reliably predicted with models trained on calcium data of day 1, with no difference in accuracy compared to within-day decoding (**Fig 3C**). These results suggest a stable representation of the behavioral state during TS by repeatedly identified neurons.

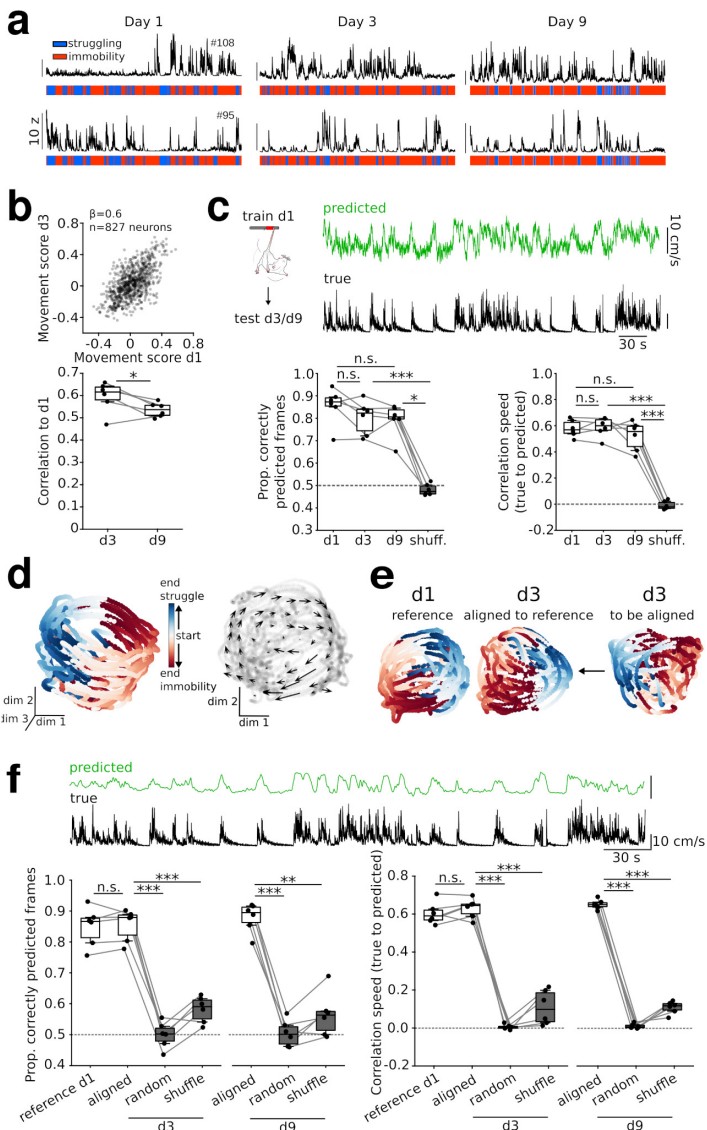

**Fig 3. Stable population coding of threat coping responses over time. (a)** Examples of an immobility-active (top) and struggling-active neuron (bottom) over recording days 1, 3, and 9. **(b)** Top: Correlation of TS movement scores on d1 and d3 (top) and mouse averages of correlations to d1, $\beta = 0.60 \pm 0.04$, $p < .001$. Bottom: correlations between days per animal, $t = 2.68$, $p = 0.044$, paired $t$ test. **(c)** Decoding TS behavior on subsequent days using models trained on calcium activity of repeatedly active neurons on d1. Top: example of predicted speed on d3. Bottom: Prediction of struggling/immobility (left, d3: vs. d1: $t = 2.07$, $p = 0.374$, vs. shuffle: $t = 13.91$, $p = 10^{-4}$, d9: vs. d1: $z = 1.92$, $p = 0.219$, vs. shuffle: $z = 2.88$, $p = 0.016$) and correlation of speed (d3: vs. d1: $t = 0.52$, $p = 1$, vs. shuffle: $t = 19.84$, $p = 2 * 10^{-5}$, d9: vs. d1: $t = 1.49$, $p = 0.787$, vs. shuffle: $t = 14.12$, $p = 10^{-4}$). **(d)** Example 3-d manifold (left) and corresponding 2-d flow field (right). **(e)** Example of across-day manifold alignment. Manifolds are constructed using all available neurons of each day. **(f)** Across-day predictions as in (**c**) but using aligned manifolds (struggling/immobility: d3: vs. reference: $t = 0.55$, $p = 1$, vs. random: $t = 21.53$, $p = 10^{-5}$, vs. shuffled: $t = 12$, $p = 2 * 10^{-4}$, d9: vs. reference: $t = 1.18$, $p = 0.872$, vs. random: $t = 18.51$, $p = 2 * 10^{-5}$, vs. shuffle: $t = 8.43$, $p = 0.001$; correlation of speed: d3: vs. reference: $t = 1.36$, $p = 0.70$, vs. random: $t = 28.66$, $p = 3 * 10^{-6}$, vs. shuffle: $t = 20.31$, $p = 2 * 10^{-5}$, d9: vs. reference: $t = 2.54$, $p = 0.155$, vs. random: $t = 52.41$, $p = 10^{-7}$, vs. shuffle: $t = 35.24$, $p = 10^{-6}$). (**b, e**) Dashed lines: chance level. One-way repeated measures ANOVAs followed by paired $t$ tests with Bonferroni correction, $n = 6$ mice. *$p < 0.05$, **$p < 0.01$, ***$p < 0.001$. The data underlying this figure can be found at https://doi.org/10.5281/zenodo.10378756.

The above analysis relied on repeatedly identified neurons that could be coregistered across multiple recording days. However, around 60% of neurons were not found to be active on all imaging days (**Fig 1D**), leaving a changing set of active neurons observable on each recording day. The partly changing set of active neurons raises the question to what extent the dynamics of the prefrontal population as a whole might remain stable over time. We thus assessed the full population of neurons, including cells that were only identified on a single recording day, during repeated TS exposure. To compare population activity of different sets of neurons across days, we extracted the low-dimensional neural manifold spanned by activity on each day using nonlinear dimensionality reduction (isomap). When visualized, population activity traversed a roughly circular manifold spanning passive and active coping (**Fig 3D** and **S2 Video**). This was consistently observed across mice (**S6 Fig**). As the manifolds for each day are constructed from partly nonoverlapping sets of neurons, they span different axes. If the structure of population activity remains stable over time (i.e., neuronal activity maps to a common manifold), a set of rotations of the low-dimensional axes will exist that matches the manifold of one day to that of another. We applied orthogonal procrustes transformation to optimally rotate the manifolds of 2 recording days ("manifold alignment"; **Fig 3E** and **S3 Video**), and compared the results to 2 control conditions: to random rotations during the alignment procedure ("random") and to aligned manifolds constructed from circularly shifted calcium traces of individual neurons so as to remove temporal correlations between individual neurons ("shuffle"). After alignment, models trained on the manifold of day 1 could predict behavior on days 3 and 9 equally well as within day 1, while decoding with random or shuffle controls yielded results close to chance level (**Fig 3F**). Similarly, based on aligned manifolds, changes in the animals' speed during TS could be reliably predicted across days (**Fig 3F**). Decoding with aligned manifolds was robust against the choice of dimensionality reduction method and the number of selected dimensions (**S7A–S7D Fig**). Across-day predictions from aligned manifolds did not depend on the activity of repeatedly identified neurons as similar results were obtained when restricting the analysis to manifolds constructed from neurons that were identified on a single recording day only (**S7E and S7F Fig**). These data jointly suggest a low-dimensional and stable geometry of prefrontal population activity during coping behavior across days.

### Preserved structure of threat-related population activity across individuals

Previous work indicated that the structure of cortical population activity is preserved across subjects performing motor and navigational tasks [12–14]. The temporally stable structure of coping-related prefrontal population activity within individuals and the consistent emergence of struggle and immobility-responsive neurons across mice thus prompted us to test whether population dynamics of individual mice can be aligned to a common manifold. Similar to the across-days analysis, we performed the alignment procedure on manifolds obtained from day 1 recordings of pairs of animals (**Fig 4A**). After alignment, the behavioral state of struggling/immobility as well as changes in speed during TS of one mouse could be reliably predicted with models trained on the manifold of another mouse, while the same was not true after random rotation during alignment or when constructing the manifold from time-shifted calcium signals (**Fig 4B and 4C**). Across-mouse predictions after alignment reached similar accuracy as within-mouse predictions (**Fig 4C**). These data jointly point to a preserved structure of population activity across individual mice.

## Discussion

We revealed a stable coding regime during repeated TS exposure in which neurons maintain comparable TS activation profiles as well as robust and specific movement tuning over time.

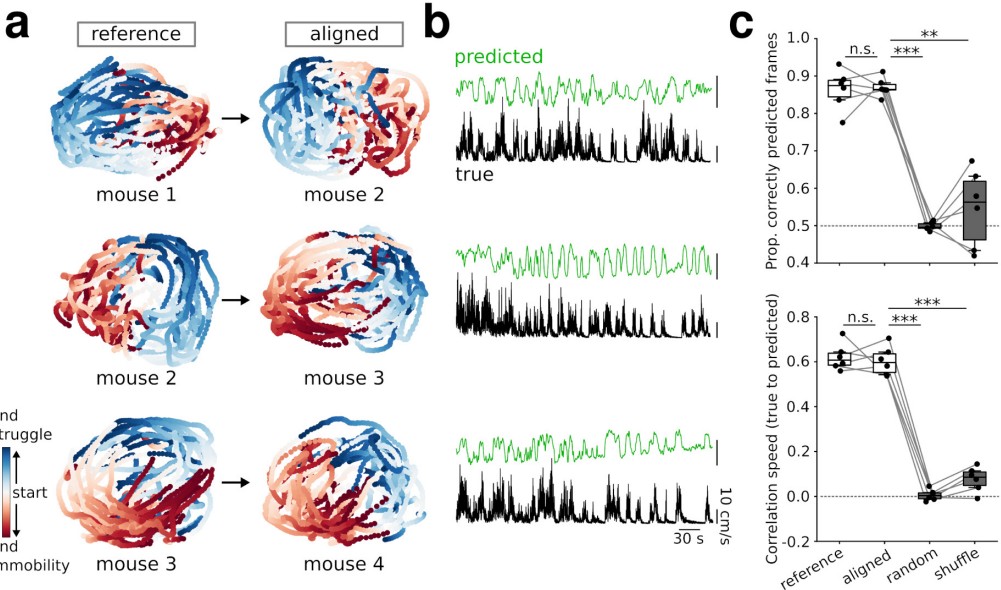

**Fig 4. Manifold structure during TS is preserved across individuals. (a)** Examples of manifolds of mice aligned to the orientation of the manifold from another individual. **(b)** Speed during TS predicted with aligned manifolds trained on the reference subject. Same pairs of mice as shown in (**a**). **(c)** Predictions of TS struggling/immobility (vs. reference (i.e., within mice): t = 0.24, p = 1, vs. random: t = 25.83, p = 4 * 10$^{-6}$, vs shuffle: t = 7.4 p = 0.002) and correlation between predicted and true speed between mice (vs. reference: t = 0.44, p = 1.0, vs. random: t = 27.96, p = 3 * 10$^{-6}$, vs. shuffle: t = 16.31, p = 4 * 10$^{-5}$). Dashed lines: chance level. One-way repeated measures ANOVAs followed by paired *t* tests with Bonferroni correction, *n* = 6 pairs of mice. ***p < 0.001. The data underlying this figure can be found at https://doi.org/10.5281/zenodo.10378756.

On the population level, we identified a low-dimensional neural manifold that was preserved across time and individuals.

mPFC neurons showed a characteristic preference for baseline or TS, with a large fraction of neurons reducing their activity when entering the TS state. This result is reminiscent of single-unit recordings from rat mPFC during single FS exposure [10]. We extend on these previous observations by providing evidence that TS selectivity remains highly correlated over a time period of 9 days. In addition to stable TS preference, we revealed that the tuning to movement within TS remained highly correlated over days. Notably, behavior of the animals changed during this time toward longer immobility duration, likely reflecting classical habituation to repeated stress [24]. This change in behavior was accompanied by opposing changes in the mean activity of struggling-active and immobility-active neurons while the neuron's tuning to movement during TS remained stable. These findings jointly point to a "rigid" encoding regime, in which any given neuron responds during the same (repeated) behavior in a stereotyped and consistent manner. They furthermore add to the ongoing debate as to whether activity in associational cortices changes over time or remains stable when animals are exposed to the same task [25–28]. Our findings are in line with the latter and support the notion that temporally stable tuning to task variables might be a common feature of the mPFC not only during cognitive tasks but also during innate responses to threat.

What might be the mechanisms of stable tuning to TS movement? In a contextual fear conditioning paradigm, shock-responsive mPFC neurons reactivate upon remote (14 days after shock encoding) but not upon recent recall (1 day after shock encoding) [29]. This finding has been interpreted in the context of systems consolidation theory to reflect the maturation of early imprinted mPFC engrams that mature over time, leading to a full reactivation of the

engram upon stimulus reexposure only after some time has passed. [29] In contrast to fear learning, neuronal responses during threat emerge from the first exposure to TS and persist in a stereotyped manner over the following days. This suggests that learning mechanisms such as synaptic plasticity among prefrontal neurons that are crucial for the acquisition of fear memory [30] might not play a dominant role in the initial emergence of TS-specific responses. Rather, movement tuning during TS might be constrained by existing shared synaptic input or by the recurrent connectivity within the mPFC network. Nevertheless, the coactivation of neurons with similar movement tuning during subsequent TS exposures might consolidate recurrent weights among them to further support the stabilization of movement tuning over successive days.

Trajectories through neural state space have been proposed to underlie cortical computations [31–33]. In line with that proposal, we showed that population dynamics occurring over days can be mapped onto a low-dimensional manifold traversed by circular trajectories between struggling and immobility states. Using a rotational alignment procedure, we show these trajectories to be confined to a common manifold across days. Similarly, population dynamics across mice are preserved in the same way. Since we observed highly consistent tuning of individual neurons to TS movement across time and individuals, it is likely that this allowed manifolds to be alignable across conditions. Our finding of a preserved structure of population activity both in time and across subjects bears similarity to and extends upon previous reports: Stable latent dynamics of population activity despite changing cell identities on different recording days have been observed in monkey motor cortex upon learning of a motor task [34]. Moreover, latent dynamics of population activity in motor cortex [35], orbitofrontal cortex [36], and hippocampus [37–39] are preserved across individuals executing learned tasks and appear in sensory-motor cortex during foraging for rewards [40]. Our results imply that stable and preserved population activity patterns are not restricted to cognitive or motor behaviors but that they also characterize the mPFC during innate stress coping.

## Methods

### Ethics statement

Experimental protocols were performed in agreement with national legislation (license G20-110, approved by the Animal Ethics Committee of the Regierungspräsidium Freiburg).

### Animals

Experiments used adult (12- to 16-week-old) Thy1-GCamp6f mice [41] (Jackson Labs #025393) of both sexes maintained on a heterozygous background by crossing with C57BI6/J mice (4 males, 2 females). The mice were housed in temperature- and humidity-controlled facilities on a 12-hour light–dark cycle (lights on: 7 AM) with ad libitum access to food and water.

### Surgical procedures

A gradient reflective index (GRIN) lens of 1 mm diameter was used (4 mm length, ProView Integrated, Inscopix) for optical access during in vivo calcium imaging. Mice were anesthetized with isoflurane (induction: 3%, maintenance: 1% to 2% in $O_2$) and placed on a heating pad in a stereotaxic frame. A craniotomy with a diameter of approximately 1 mm was performed over the mPFC (coordinates: A/P 1.7 mm, M/L 0.6 mm, 5° angle). To assist penetration of the lens, a track was made to the final D/V depth (1.4 mm from the brain surface) using a G21 injection needle. The lens was then slowly lowered into the mPFC. The craniotomy was

sealed off with Vaseline, and the lens cemented to the skull using Superbond C&B (Sun Medical). Buprenorphine (0.1 mm/kg body weight) and Carprofen (0.1 mg/kg body weight) were injected SC prior to and for 2 days after surgery (2 to 3 injections of buprenorphine and 1 injection of carprofen/day). Buprenorphine was additionally supplied for 2 days in the drinking water overnight (10 mg/l). The animals were daily habituated to the experimenter, experimental room, and to wearing a dummy microendoscope for at least 1 week prior to behavioral experiments.

## Miniscope imaging

On each day of imaging, mice were habituated in their homecage with the microendoscope attached to the head for approximately 10 minutes before the beginning of the experiment. Imaging acquisition was done with an nVista DAQ box (Inscopix) at a frame rate of 20 Hz (gain 2–5, LED power 1.0 to 1.2 mW/mm$^2$) using Inscopix software (Data Acquisition Software). Calcium imaging and behavioral video recordings were synchronized by triggering a blue LED in the field of view of an overhead behavioral camera for offline alignment with the calcium data. Imaging was performed during baseline and TS sessions on days 1, 2, 3, and 9. Due to technical issues with some animals on individual recording days (e.g., field-of-view shift), analyzable data could not be obtained for all mice on all recording days. For all analysis, we selected a subset that (1) contained recordings from the first day of TS exposure and (2) maximized the number of mice and the covered days, which resulted in the 6 mice with imaging on days 1, 3, and 9 in the final manuscript.

## Behavior

The behavioral experiments were performed between 2:00 PM and 5 PM. Mice were initially recorded for 10 minutes in their home cage (baseline). Subsequently, TS was performed by fixing the tail on a horizontal bar (raised about 20 to 30 cm). The behaviors were recorded at 20 Hz using an overhead camera (LifeCam Studio, Microsoft). Some TS recordings were additionally recorded from the side (C505 HD Webcam, Logitech).

## Behavioral classification and quantification

TS behavior (coping style) of the animal was manually labeled as "struggling" and "immobility" based on video recordings. We did not label frames when the coping style was ambiguous. The position of the animal during baseline and TS was tracked with DeepLabCut [42], allowing quantification of the speed for various body parts (baseline: nose, left hip, right hip, basetail; TS: camera, neck, left foot, right foot). For quantification, speed was averaged between all body parts.

## Histological processing

Mice were deeply anesthetized by intraperitoneal injection of ketamine/xylazine (100/13 mg/kg body weight). After cessation of pain reflexes, the thorax was opened to expose the heart, and an incision was made into the right atrium. Perfusion commenced into the left ventricle with ice-cold phosphate-buffered saline (PBS, approximately 5 ml), followed by 4% paraformaldehyde (PFA, approximately 15 to 20 ml). The brains were postfixed in 4% PFA at 4°C for 2 days. Coronal sections were cut with a vibratome (Leica VT1000S, 100 μm thickness), stained with 4′-6-diamidino-2-phenylindole (DAPI, 1:1,000 in PBS), and mounted on microscope slides with Mowiol. Confocal image stacks were obtained with a confocal laser-scanning microscope (Zeiss LSM 900). For immunohistochemistry, the brains were postfixed for 1 day,

and coronal sections were cut to 70 μm thickness. The sections were washed 3 times in PBS and preincubated (PBS with 10% NGS and 0.3% Triton X-100) for 60 minutes at room temperature. Subsequently, the brain slices were incubated with primary antibody diluted in 5% NGS, 0.3% Triton X-100 and PBS, overnight at 4°C in the dark. The primary antibodies used in this study were rabbit polyclonal anti-EMX1 (EMX; 1/100 dilution, Thermo Fisher, Cat#PA5-52294) to label cortical pyramidal cells [43], rabbit anti-parvalbumin (PV; 1/500 dilution, Swant, Cat#PV 27), and rabbit polyclonal anti-somatostatin-14 (SOM; 1/500 dilution, Peninsula Laboratories, Cat#T-4102) to label GABAergic interneuron populations. On the following day, the sections were washed 3 times with PBS and incubated for 2 hours in room temperature with secondary antibody (dilution 1/1,000, goat polyclonal anti-rabbit CY3, Jackson ImmunoResearch, Cat#111-165-003) diluted in PBS, 3% NGS, and 0.3% Triton X-100.

## Analysis of confocal images

To characterize the GCaMP-positive neurons in mPFC, we analyzed confocal image stacks of prelimbic and cingulate cortex acquired at 20 or 60× magnification. First, GCaMP-, SOM-, PV-, and Emx1-positive neurons were identified and segmented in 3D with CellPose 2.0 [44]. For each identified GCaMP-positive neuron, we quantified whether it was positive for the interneuron markers SOM or PV, or the pyramidal neuron marker EMX, by measuring spatial overlaps. We considered overlap if the 3D spatial position of the GCaMP ROI contained ROI from the other channel with at least 10% of the same size. We also quantified overlap the other way around, with SOM, PV, or EMX as reference. To characterize the morphology of the GCaMP neurons, we reconstructed the apical dendrites using Simple Neurite tracer [45] in ImageJ. For each identified soma, we traced the main branch of the dendrites until it could no longer be separated from other dendrites.

## Extraction of calcium signals

The microendoscopic videos were cropped to the smallest region containing only nondistorted neurons. Motion correction and calcium trace extraction were achieved using CaImAn (CNMF-E [46]) with the following parameters: gSig_filt = 10, rf = 120, stride_cnfm = 40, gSig = 10 gSiz = 41, ring_size_factor = 1.5, min_corr = 0.8, min_pnr = 8. We used a custom GUI to discard low-quality components and to merge components coming from the same source. In order to identify the same neurons in different imaging sessions, we aligned the field of views and identified overlapping neurons with the CellReg method [47]. This method models the distribution of spatial correlations and centroid distances between neighboring pairs of components identified from different recording sessions. We only considered components with a max distance <20 μm as candidates to be the same neurons. As a trade-off between false-positive and false-negative, we used a registration threshold of 0.5 to assign components to the same neuron. After registration of neurons, the extracted calcium traces were corrected for slow drifts from the baseline using a running percentile filer (10th percentile, window size 30 seconds). Finally, the traces were z-scored before analysis. To complement the z-scored calcium traces, we computed transients with a mask detecting whether the calcium signal at each frame increased by more than 2 standard deviations. This provided a boolean array with the same length as the normal calcium trace.

## TS selectivity score

To quantify TS selectivity, the baseline and TS sessions were concatenated and scaled to range from 0 to 1 for each neuron. TS selectivity was then defined as the difference in the mean

scaled calcium signal during TS and baseline divided by the sum of both means. We classified neurons with TS selectivity score below −0.2 as baseline selective and above 0.2 as TS selective.

## Movement tuning

Tuning to movement during baseline recording and to struggling/immobility (TS movement) was quantified as the Pearson's correlation between the speed of the animal and the calcium signal for each neuron. We defined struggle- and immobility-selective neurons as neurons with correlation above 0.2 or below −0.2, respectively.

## Temporal characterization of struggle and immobility neurons

The temporal relationship between neuronal activity and coping (struggle and immobility) onsets were examined by performing a cross-correlation procedure. We focused our analysis on struggle neurons during struggling behavior, and immobility neurons during immobility behavior. We acquired peri-event time histograms (PETHs) aligned to the coping onset events (0.5 seconds prior to 0.5 seconds after the onsets). For each event, we normalized the neuronal data and behavioral speed before calculating average responses per neuron. Cross-correlations were performed between the averaged responses of neurons and the behavioral speed. From these correlations, the time lags were estimated as the time point of maximal and minimal correlation for struggle and immobility neurons, respectively. We considered a neuron to have significant time lag if the correlation value at the time lag were above 1 standard deviation of the distribution of correlations of all analyzed neurons of a given coping type.

## Decoding behavioral context from neurons

For decoding of behavioral context (baseline versus TS), we balanced the data by removing random frames from the baseline recording until the total duration was equal to TS. Then, using a 10-fold cross validation protocol, we used logistic regression (C = 0.1) to fit a model and predict from neuronal activity. In addition, we tested a support vector machine (C = 1), Gaussian naïve bayes, and ridge classifier (alpha = 50). The performance of the decoder was estimated using average accuracy classification score for each fold. This and all subsequent decoding algorithms from this study were implemented using the *scikit-learn* package [46] (https://scikit-learn.org). As shuffle control, we independently circularly shuffled the activity of each neuron before generating train and test data for fitting and prediction. We used the same indices for real and shuffled data. This shuffle control procedure was used for all subsequent decoders trained on data from neurons.

## Decoding coping style from neurons

To ensure balanced training data, we first took all frames of the coping style (struggling or immobility) with the longest total duration and removed random frames until both copying styles had equal total duration. All remaining frames were used for training and prediction following a 10-fold cross validation protocol. The accuracy scores for each fold were averaged to give an estimate of model performance. Various decoding algorithms were used, including logistic regression (LogisticRegression; C = 0.01; main figures), linear support vector machine (LinearSVC; C = 1, kernel = "linear"), ridge classifier (RidgeClassifier; alpha = 50), and Gaussian naïve bayes (GaussianNB). The data were standardized before applying the models.

### Decoding behavioral speed from neurons

The behavioral speed during TS and baseline were decoded using ridge regression (alpha = 50). We first removed the last part of the baseline data in order to ensure equal length between the recordings. Then, following a 10-fold cross validation protocol, we extracted the same indices for baseline and TS (both real and shuffled data) to be used as training and test data for each fold. To quantify the performance of the decoder, we calculated Pearson's correlation between the predicted and true speed for each fold before averaging the performances separately. We used all combinations of training and test data in order to compare the performances both within and between TS and baseline. The data were standardized before applying the models.

### Across days decoding from neurons

Only neurons that were detected on all 3 recording days (See section Extraction of calcium signals) were used for across days decoding. For across days binary classification of behavioral context (see section Decoding coping style from neurons), we balanced the data by removing random frames from the baseline recording until the total duration was equal to TS and ensured equal number of frames between the recording days. For both binary classification of behavioral context and speed decoding (see section Decoding behavioral speed from neurons), we trained a model with the data from day 1 and tested the model on the data of all 3 recording days. Following a 10-fold cross validation protocol, we selected the same indices for all dataset for each of the 10-folds.

### Linear modeling

For linear model analyses, the entire baseline or TS duration was used for all neurons with preferred baseline (TS selectivity score $<-0.2$, $n = 277$) or TS activity (TS selectivity score $>0.2$, $n = 160$), respectively. To assess to what extent calcium signals of individual neurons can be explained by ongoing movement, we used a linear regression model taking the continuous movement signal as a single variable. To assess the explanatory power of the activity of other neurons, a separate linear regression model was used which took the activity of 50 randomly selected neurons as variables to explain the activity of the target neuron. Explained variance (EV) between the predicted and true calcium signal was obtained in a 10-fold cross-validation procedure by averaging EV values of each fold. To test whether each neuron's activity can be significantly explained by movement, the movement signal was randomly time-shifted (20 seconds to 2 minutes, 1,000 iterations). EV was computed for each of the time-shifted cases. A neuron was considered significant if the EV of the actual movement trace exceeded the 95th percentile of the EVs obtained from the surrogate distribution.

### Manifold embedding

We first convolved the calcium traces with a Gaussian kernel of σ = 250 m*s* and feature-scaled the traces between 0 and 1 using a quantile transformer (n_quantiles = 100). These traces were embedded into a 5-dimensional space using the nonlinear dimensionality reduction method *isomap* (n-neighbors = 500, metric = euclidean). To save computation time, we fitted the model with 60% temporally downsampled data and used this model to embed the full dataset. We also tested other dimensionality reduction methods, including spectral embedding (n_neighbors = 1,000, metric = euclidean) and PCA. For PCA and spectral embedding, we did not need to downsample the fitting data. All dimensionality reduction was performed using the *scikit-learn* package.

## Visualization and estimation of manifold trajectories

Visual inspection of the manifolds revealed consistent rotational trajectories between struggle and immobility epochs. The rotational trajectories were usually spanned by the first 2 dimensional axes but were occasionally spanned by other axes in 5-dimensional space. We further characterized this rotational trajectory by selecting corresponding 2 axes after visual inspection. For visualization (Figs 3D, 3E, and 4A), we color coded the manifolds according to the relative beginning and ending of each period of struggling and immobility. For each period, we normalized an array of the length (0,1,2, . . . n frames) between 0 and 1 with a min-max scaler. These normalized traces were concatenated to match the calcium traces and used as color code for the plot. To visualize the flow field on the manifold trajectories, we calculated velocity vectors between consecutive time points, spatially binned ($15 \times 15$ bins) the 2-dimensional subspace, and averaged all velocity vectors inside the same bin. These vectors were visualized with a quiver plot using the *matplotlib* library. We also estimated the trajectory direction using an angular approach. First, the data points on the manifold were temporally binned to 2.67 Hz (800 bins). We then recalculated the velocity vectors on the time-binned manifold and measured the angle between velocity vectors $t$ and $t$-1 at time point $t$:

$$(\theta_t - \theta_{t-1})mod(2\pi),$$

where $\theta$ is the angle of the velocity vector given by arctan2. Between 0 and $\pi$ radian correspond to leftward turns, while between $\pi$ and $2\pi$ radian correspond to rightward turns.

## Manifold alignment

Each manifold was first translated into a common reference frame by shifting its n-dimensional mean centroid into its origin. Next, the manifolds were scaled into unit-length arrays by dividing each element in the arrays by its norm. To find the best possible alignment transformations, we generated "template manifolds" by computing an average epoch of struggle and immobility from the full manifolds. The behavioral epochs were scaled between 1 and 20, and digitized, such that a behavioral epoch starts at 1 (beginning of struggle/immobility) and ends at 20 (end of struggle/immobility) with evenly spaced digits. The manifolds were averaged by taking the mean of each digit of each dimension. All template manifolds had shape $5 \times 40$ (manifold dimensions $\times$ epoch length). To align 2 template manifolds A and B, we kept one template manifold as reference and aligned the other with orthogonal procrustes transformation. Given a matrix $A_{mxn}$ and a reference matrix $B_{mxn}$, we were searching for an orthogonal transformation matrix $Q_{nxn}$, which makes AQ as close as possible to B:

$$\arg \min \|AQ - B\|_F^2,$$

where $\|\cdot\|_F$ denotes the Frobenius norm. The optimal solution is obtained using singular value decomposition (SVD),

$$A^T B = U\Sigma V^T$$

$$Q = UV^T,$$

with singular values listed in decreasing order. To align the full manifolds, the optimal orthogonal transformations Q from the template manifolds were applied to a full manifold of A, M,

such that

$$alignedmanifold = MQ.$$

To compare the alignments to chance, we used 2 different approaches. The first approach was to generate a random rotation matrix $Q_{random}$ and apply it to the full manifold M, in which

$$randomrotatedmanifold = MQ_{random}.$$

The random rotation matrix was drawn from the Haar distribution with a determinant of 1, using *special_ortho_group* from the python library *scipy.stats*. This procedure was repeated 100 times to estimate the performance of an average random rotation. As a second approach, we independently circle shuffled the activity of each neuron before performing dimensionality reduction and alignment on the reference manifold. For all alignments, the reference manifold was always kept constant.

## Cross-manifold decoding

To assess whether the low-dimensional population dynamics were conserved over different days and mice, we utilized a decoding approach in which we predicted behavior between different manifolds. We trained a model on the relationship between a manifold and the behavior corresponding to the same animal and day and predicted behavior from a manifold of a different animal or day. We compared the decoding scores of 2 aligned manifolds with that of 2 manifolds in which one had been randomly rotated, or the neural activity had been shuffled before the embedding (see section Manifold alignment for a description). As models for decoding, we used logistic regression (C = 0.01) to decode struggle/immobility (similar approach as described in section Decoding coping style from neurons), and ridge regression (alpha = 50) to decode speed (similar approach as described in section Decoding behavioral speed from neurons). To ensure balanced datasets for the binary classification, we extracted an equal number of struggle and immobility frames within and between the 2 datasets. Using 10-fold cross-validation, the same indices were selected for both datasets, including alignment, random rotations, and shuffle. For cross-animal decoding, we always aligned and decoded between the manifolds in a circular manner. For each animal in the list (animal$_1$, animal$_2$ . . . animal$_6$), we trained a model from the manifold of animal$_{n-1}$ and did alignment (or random rotations/shuffle) and prediction with the manifold of animal$_n$.

## Data analysis and statistics

Data analysis and statistical tests were performed in Python using custom written scripts. Open source toolboxes included Numpy, SciPy, Matplotlib, Scikit-learn, Math, Statsmodels, and h5py. Linear mixed effects model (Statsmodels) was used for correlation analysis (Figs 1D, 1F, 2B, and 3B) based on neural data pooled from different animals. We did this to account for potential differences between animals. We treated time points as fixed effects and fit random effects (slope and intercept) using the animal ID. We reported the standardized coefficients (ß), standard error of ß, and *P* values for the fixed effects. For all group comparisons, normality of data was determined with Shapiro–Wilk tests. Data violating the assumption of normality were assessed using nonparametric tests. Comparisons between 2 dependent groups were performed with paired *t* tests or the nonparametric Wilcoxon signed-rank tests, while comparison between 2 independent groups were performed with independent *t* tests or the nonparametric Mann–Whitney U test. To compare more than 2 dependent groups, we used one-way

repeated measures ANOVA. The resulting *p*-values were corrected for multiple comparisons with Bonferroni's method. All statistical tests were 2-sided.

## Supporting information

**S1 Video. Simultaneous acquisition of TS behavior and calcium data.** Top: Extraction of motion speed (left) and binary periods of struggling/immobility (right) during TS. Bottom: Raw field-of-view of GCaMP6f signal (left), df/f-converted calcium signals (middle) and calcium activity of isolated components extracted with CNMF-E.
(MP4)

**S2 Video. Temporal evolution of the neural manifold during TS.** Projection of the manifold obtained from nonlinear dimensionality reduction. Struggling is shown in blue, immobility in red, with hue value indicating the relative progression through each individual epoch of struggling/immobility. Video shows 2× speed.
(MP4)

**S3 Video. Illustration of the manifold alignment procedure.**
(MP4)

**S1 Fig. Histological verification of lens locations in the mPFC and immunohistochemical and morphological characterization of GCaMP6f-expressing neurons. (a)** Histological verification of recording sites (1 mm lenses) in the prelimbic (PrL) and cingulate cortex (Cg1). M2: secondary motor cortex, IL: infralimbic cortex, DP: dorsal peduncular cortex. Yellow lines: estimated imaging planes. (**b**) Percentage of neurons recorded in different brain regions. (**c**) Zoomed-in view of a z-projected confocal image stack below the imaging lens. (**d**) GCaMP6f-positive neurons display spiny dendrites. Inset: Enlarged views with spines indicated by arrows. (**e**) Confocal z-projection of GCaMP6f-positive neurons (left) and reconstructed apical dendrites from the same field of view (right) showing characteristic dendritic morphologies of pyramidal neurons. (**f**) Measurement of the colocalization of GCaMP6f with markers of pyramidal cells (Emx1, *n* = 1 mouse) and GABAergic interneurons (parvalbumin (PV) and somatostatin (SOM), *n* = 3 mice). Examples of immunostainings (left) and neurons that were positive for GCaMP6f and/or one of the marker proteins (right). Positive neurons were detected with the CellPose algorithm [44]. (**g**) Top: Quantification of the percentage of marker-positive cells that coexpress GCaMP6f (*n* = 1,869 Emx1-positive neurons, *n* = 412 PV-positive neurons, *n* = 377 SOM-positive neurons tested). A large fraction of Emx1-positive pyramidal cells coexpressed GCaMP6f while expression was rare in interneurons. Bottom: Percentage of GCaMP6f-positive neurons expressing Emx1, PV or SOM (*n* = 708, 2,412, and 2,787 GCaMP6f-positive cells tested). The vast majority of GCaMP6f-positive neurons expressed Emx1 while expression of interneuron markers was rare. Dots show mouse averages. The data underlying this figure can be found at https://doi.org/10.5281/zenodo.10378756.
(EPS)

**S2 Fig. Examples and metrics of cell registration across days. (a)** Example field-of-view of all neurons detected on each individual day and their overlap with neurons detected on all other days. Inset: Quantification of true positive (left) and true negative cell detection scores obtained with the CellReg algorithm [47]. Data points are mouse averages. (**b**) Example of stable spatial locations in the field of view and stable response properties. The ROIs shown are classified as struggle-selective neurons (see Fig 3). A short segment of calcium activity during TS is shown for 2 of the cells. Both neurons maintain their struggle-related activity over days.

The data underlying this figure can be found at https://doi.org/10.5281/zenodo.10378756.
(EPS)

**S3 Fig. Additional analyses of TS selectivity and prediction of baseline and TS state with different decoding models. (a)** TS selectivity scores (as shown in Fig 1e) plotted separately for prelimbic (PrL, $n$ = 704 neurons) and cingulate (Cg, $n$ = 126 neurons) areas. t = 2.96, p = 0.003, unpaired $t$ test. (**b**) Neurons of both brain regions show comparable correlation in their TS selectivity on days 1 and 9 (PrL: Pearson's r = 0.68, p = 3 * $10^{-97}$, Cg: r = 0.60, p = $10^{-13}$). (**c**) TS selectivity scores plotted separately for female ($n$ = 275 neurons from 2 mice) and male mice ($n$ = 552 neurons from 4 mice). There was no difference between both sexes (t = 0.61, p = 0.544, unpaired $t$ test). (**d**) Neurons of both sexes show comparable correlation in their TS selectivity on days 1 and 9 (females: Pearson's r = 0.69, p = $10^{-40}$, males: r = 0.69, p = 4 * $10^{-80}$). (**e**) Prediction of baseline vs. TS state with different decoding models. Left: Using a support vector machine, accuracy across days was lower compared to within-d1 (d3: t = 4.93, p = 0.021, d9: t = 6.36, p = 0.007) but above chance level (vs. shuffled: d1: t = 224.22, p = $10^{-10}$, d3: t = 31.41, p = $10^{-6}$, d9: t = 33.36, p = $10^{-6}$). Middle: Same comparison between days (d3: t = 3.1, p = 0.134, d9: t = 3.49, p = 0.087) and shuffle (d1: t = 43.83, p = $10^{-7}$, d3: t = 36.84, p = $10^{-6}$, d9: t = 25.36, p = $10^{-6}$) using Gaussian naive bayes classifier. Right: Across days (d3: t = 4.7, p = 0.027, d9: t = 9.54, p = 0.001) and shuffle (vs. shuffled: d1: t = 107.74, p = $10^{-9}$, d3: t = 27.74, p = $10^{-6}$, d9: t = 49.49, p = $10^{-7}$) comparisons using a ridge classifier. One-way repeated measures ANOVAs followed by paired $t$ tests with Bonferroni correction, $n$ = 6 mice. *p < 0.05, **p < 0.01, ***p < 0.001. The data underlying this figure can be found at https://doi.org/10.5281/zenodo.10378756.
(EPS)

**S4 Fig. Additional analyses of behavioral state prediction during TS, activity modeling, and neuronal responses during struggling and immobility. (a-c)** Decoding binary states of struggle and immobility does not depend on the chosen model (predictions vs. shuffle: (**a**) t = 10.7, p = $10^{-4}$, (**b**) t = 20, p = 5 * $10^{-6}$, (**c**) z = 2.88, p = 0.004). Paired $t$ tests, $n$ = 6 mice. (**d**) Linear model predictions of calcium signals of individual neurons with the calcium traces of 50 randomly selected other neurons. Black: true signals, colored: predicted signals. EV, explained variance. (**e**) Cumulative histogram of explained variance for baseline ($n$ = 277 cells) and TS ($n$ = 160 cells). (**f**) Mouse averages of the data shown in (**e**). t = −8.83, p = 3 * $10^{-4}$, paired $t$ test. $n$ = 6 mice. (**g**) Analysis of calcium activity aligned with behavioral transitions from movement to immobility during TS. Top left: Trial-averaged movement and calcium signal (mean ± SEM) of a neuron that significantly elevates its activity before behavioral transition ("early" response, magenta) and a cell with increasing activity following behavioral change ("late" response, blue). Bottom: Proportion of early, late, and cells without significant modulation of their activity around transition points (gray). Right: Summary of averaged calcium activities around behavioral transitions of all significantly modulated neurons. (**h**) Same as (**g**) but for transitions from immobility to struggling. (**i**) Analysis of movement tracking with different body parts. Top left: Schematic of the different tracked parts during baseline and TS. In the main text, movement correlations and predictions of movements are based on the averaged speed of all measured body parts. Top middle: Mean speeds for all body parts. Top right: Mean correlation in speed between parts. Bottom: Significant prediction of baseline (left) and TS speed (right) of individual body parts compared to shuffled data (from left to right: t = 9.66, p = 2 * $10^{-4}$, t = 8.06, p = 5 * $10^{-4}$, t = 7.09, p = 9 * $10^{-4}$, t = 10.66, p = $10^{-4}$, t = 11.08, p = $10^{-4}$, t = 10.47, $10^{-4}$, paired $t$ tests, $n$ = 6 mice). ***p < 0.001. The data underlying this figure can be found at https://doi.org/10.5281/zenodo.10378756.
(EPS)

**S5 Fig. Additional analyses of the stability of struggling- and immobility-related responses.** (**a**) Increasing proportion of struggling-selective neurons (TS movement score >0.2, left, d1 vs. d3: t = −3.71, p = 0.042, d1 vs. d9: t = −3.63, p = 0.045, d3 vs. d9: t = −0.79, p = 1) and constant proportion of immobility-selective neurons over days (TS movement score <−0.2, right, F = 0.18, p = 0.839, $n$ = 6 mice, one-way repeated measures ANOVAs followed by paired $t$ tests with Bonferroni correction). (**b**) Increasing mean activity of struggling-selective neurons during struggle epochs (left, d1 vs. d3: t = −9.43, p = $7 * 10^{-4}$, d1 vs. d9: t = −4.50, p = 0.019, d3 vs. d9: t = −1.34, p = 0.706) and decreasing mean activity of immobility-selective neurons during immobile epochs over days (right, d1 vs. d3: t = 7.88 p = 0.002, d3 vs. d9: z = 2.08, p = 0.011, d3 vs. d9: z = −0.80, p = 1, n = 6 mice, one-way repeated measures ANOVAs followed by paired $t$ tests/Wilcoxon tests with Bonferroni correction). (**c**) Correlation of TS movement score obtained separately for the first and second half of a single recording session (day 1). ß = 0.61 ± 0.06, p < 0.001, LME. (**d**) High correlation in movement scores across days was observed when neurons recorded in females (left, r = 0.57, p = $10^{-25}$, $n$ = 275 neurons) or males (right, r = 0.61, p = $10^{-60}$, $n$ = 552 neurons) were analyzed. (**e**) Similar TS movement score correlation over days for neurons recorded in the prelimbic (PrL, Pearson's r = 0.61, p = $10^{-72}$, $n$ = 704 neurons) and cingulate cortex (Cg, Pearson's r = 0.58, p = $10^{-13}$, $n$ = 126 neurons). The data underlying this figure can be found at https://doi.org/10.5281/zenodo.10378756.
(EPS)

**S6 Fig. Manifold structure of individual mice.** Left graphs: 2-dimensional trajectories of all struggle and immobility epochs during d1 of TS, color coded by relative start and end times. Middle graphs: Flow fields overlaid on the same figures. Right graphs: Angles between consecutive velocity vectors on the same 2-dimensional space illustrating that population activity follows a consistent rotational trajectory in each animal. The data underlying this figure can be found at https://doi.org/10.5281/zenodo.10378756.
(EPS)

**S7 Fig. Across-day predictions with manifolds constructed with other dimensionality reduction methods and from single day-active neurons.** (**a**) Example manifold alignments using principal component analysis (PCA, left) and spectral embedding (right). (**b**) Aligned manifolds based on PCA predict TS struggle/immobility (d3: vs. reference: z = 0, p = 1, vs. random: t = 16.2, p = $5 * 10^{-5}$, vs. shuffled: t = 14.09, p = $9 * 10^{-5}$, d9: vs. reference: t = 1.12, p = 0.787, vs. random: t = 16.94, p = $4 * 10^{-5}$, vs. shuffle: t = 17.41, p = $3 * 10^{-5}$) and change in speed with comparable precision to the within-day reference (d3: vs. reference: t = 1.51, p = 0.571, vs. random: t = 39.07, p = $6 * 10^{-7}$, vs. shuffled: t = 10.51, p = 0.0005, d9: vs. reference: t = 1.52, p = 0.564, vs. random: t = 98.31, p = $6 * 10^{-9}$, vs. shuffle: t = 21.26, p = $10^{-5}$). (**c**) Same as (**b**) but for spectral embedding used to create manifolds (struggle/immobility prediction: d3: vs. reference: t = 0.49, p = 1, vs. random: t = 19.1, p = $2 * 10^{-5}$, vs. shuffled: t = 17.71, p = $3 * 10^{-5}$, d9: vs. reference: t = 0.49, p = 1, vs. random: t = 27.63, p = $3 * 10^{-6}$, vs. shuffle: t = 10.85, p = $4 * 10^{-4}$; change in speed: d3: vs. reference: t = 0.89, p = 1, vs. random: t = 24.23, p = $6 * 10^{-6}$ vs. shuffled: t = 20.07, p = $2 * 10^{-5}$, d9: vs. reference: t = 1.03, p = 1, vs. random: z = 2.88, p = 0.012, vs. shuffle: t = 12.45, p = $2 * 10^{-4}$). (**d**) Performance of Isomap embedding is robust against the number of dimensions used when predicting struggle/immobility ($p$ = 0.694) and change in speed ($p$ = 0.51). (**e**) Frame-by-frame prediction of TS struggling/immobility predicted from aligned manifolds of neurons found active only on a single recording day trained on d1 (left, d3: vs. reference: t = 0.69, p = 1, vs. random: t = 12.28, p = 0.0002, vs. shuffled: t = 6, p = 0.006, d9: vs. reference: t = 0.69, p = 1, vs. random: t = 10.77, p = 3 *

$10^{-4}$, vs. shuffle: t = 11.62, p = 3 * $10^{-4}$). (**f**) Same as (**e**) but predicting change in speed (d3: vs. reference: t = 2.17, p = 0.247, vs. random: t = 20.29, p = 2 * $10^{-5}$, vs. shuffle: t = 11.17, p = 3 * $10^{-4}$, d9: vs. reference: t = 2.01, p = 0.302, vs. random: t = 37.54, p = 7 * $10^{-7}$, vs. shuffle: t = 17.62, p = 3 * $10^{-5}$). Dashed lines: chance level. One-way repeated measures ANOVAs followed by paired *t* tests or Wilcoxon rank sum tests with Bonferroni correction, *n* = 6 mice. *p < 0.05. **p < 0.01, ***p < 0.001. The data underlying this figure can be found at https://doi.org/10.5281/zenodo.10378756.
(EPS)

## Acknowledgments

We thank Kerstin Semmler and SeongHee Cho for technical assistance.

## Author Contributions

**Conceptualization:** Ole Christian Sylte, Hannah Muysers, Marlene Bartos, Jonas-Frederic Sauer.

**Data curation:** Ole Christian Sylte.

**Formal analysis:** Ole Christian Sylte.

**Funding acquisition:** Marlene Bartos, Jonas-Frederic Sauer.

**Investigation:** Ole Christian Sylte, Hannah Muysers.

**Methodology:** Ole Christian Sylte, Hannah Muysers, Hung-Ling Chen.

**Software:** Ole Christian Sylte, Hung-Ling Chen.

**Supervision:** Jonas-Frederic Sauer.

**Visualization:** Ole Christian Sylte, Jonas-Frederic Sauer.

**Writing – original draft:** Ole Christian Sylte, Jonas-Frederic Sauer.

**Writing – review & editing:** Hannah Muysers, Hung-Ling Chen, Marlene Bartos.

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
