## [Editor Report · Decision Letter 0]

6 Jul 2023

Dear Dr Sauer, 

Thank you for submitting your manuscript entitled "Time-invariant prefrontal activity patterns during repeated exposure to intense threat" for consideration as a Short Reports by PLOS Biology.

Your manuscript has now been evaluated by the PLOS Biology editorial staff as well as by an academic editor with relevant expertise and I am writing to let you know that we would like to send your submission out for external peer review.

Once your full submission is complete, your paper will undergo a series of checks in preparation for peer review. After your manuscript has passed the checks it will be sent out for review. To provide the metadata for your submission, please Login to Editorial Manager (https://www.editorialmanager.com/pbiology) within two working days, i.e. by Jul 08 2023 11:59PM.

Kind regards,

Christian

Christian Schnell, PhD

Senior Editor

PLOS Biology

cschnell@plos.org

---

## [Decision Letter · Decision Letter 1]

8 Aug 2023

Dear Dr Sauer,

Thank you for your patience while your manuscript "Time-invariant prefrontal activity patterns during repeated exposure to intense threat" was peer-reviewed at PLOS Biology. It has now been evaluated by the PLOS Biology editors, an Academic Editor with relevant expertise, and by several independent reviewers. 

In light of the reviews, which you will find at the end of this email, we would like to invite you to revise the work to thoroughly address the reviewers' reports.

As you will see below, the reviewers the reviewers think that the study is very well executed and provides important insights. However, they raise concerns about some of the analyses, a couple of technical aspects, lack of methodological details, and some parts that are not sufficiently analysed/discussed.

Given the extent of revision needed, we cannot make a decision about publication until we have seen the revised manuscript and your response to the reviewers' comments. Your revised manuscript is likely to be sent for further evaluation by all or a subset of the reviewers.

**IMPORTANT - SUBMITTING YOUR REVISION**

*Re-submission Checklist*

*Published Peer Review*

*PLOS Data Policy*

*Blot and Gel Data Policy*

Kind regards,

Christian

Christian Schnell, PhD

Senior Editor

PLOS Biology

cschnell@plos.org

REVIEWS:

Reviewer #1: The authors of the manuscript of "Time-invariant prefrontal activity patterns during repeated exposure to intense threat" describe the neuronal activity patterns within the mPFC during an intense threat situation (Tail Suspension Task). They record Ca2+ transients before and during repeated sessions and identify common population dynamics across TS sessions. The experiments are well described and analyzed. Overall, I support publication with a few minor changes, a number of suggestions and optional items are listed below: 

* Fig 1e-h. It is informative to show the neuronal selectivity for TS not only for the population but based on the classification (TS or BS selective in Fig 1e.). For example color labels from Fig 1e, (right) onto Fig 1f. and/or running the anaylsis of Fig 1g and 1h only for the categories of Fig 1e, (right). My understanding is that results in Fig g and h include all 827 neurons that have been identified across days and not the selective neurons only. An analysis that helps to stratify individual trajectories is welcome. 

* While not critical the term "repeatedly active neurons" is misleading and per se not precise, finding a suitable term is not trivial, but something along the lines of "repeatedly identified neuron". Identifying the same neurons across sessions is not as precise as we may wish (and is not only linked to activity), despite the authors using state of the art and trying their best to reduce false positive and false negative results it would be welcomed if the authors could show metrics/results regarding this procedure in the supplementary figures.

* In line 156 the authors state "…and that this information is specific for coping behaviors". The use of "specific" is misleading as the authors cannot show specificity beyond the behavioural states that have been tested (TS mobile/immobile & baseline). 

* The authors if possible should link the role of existing mPFC evidence/literature regarding the PFC control of innate behaviours responses stronger in either the discussion or the introduction. 

While not necessary, I would be interested if the authors attempted, have data or are able to show changes of mPFC activity patterns prior to mobile or immobility phases of the TS (in contrast to BS). If so, the authors could potentially support a more mechanistic interpretation of the mPFC involvement in guiding the coping mechanisms. 

Also not necessary, and not trivial to implement, but supportive for the interpretation could be manifold representations for the BS that resemble time-, and motor activity (movement/no-movement) wise, episodes of the TS. e.g. identify BS movement trajectories and replicate the analysis. As it is not clear what are the latent causes for the current consistency within the representations. 

Reviewer #2: In this interesting study, Style et al used 1-photon calcium imaging in mice to probe the activity of prefrontal cells during repeated exposure to intense threat in a tail suspension (TS) paradigm. They showed a stable coding regime during repeated TS exposure in which neurons maintain comparable TS activation profiles as well as robust and specific movement tuning over time. Furthermore, they identified a low-dimensional neural manifold which was preserved across time and individuals. In all, the manuscript is well-written, and the data are presented orderly. The data can contribute to disentangling the prefrontal networks in controlling coping styles during threat.

I have the following suggestions and recommendations:

1) The "habituation" was defined as "a form of simple, nonassociative learning in which the magnitude of the response to a specific stimulus decreases with repeated exposure to that stimulus"[1,2]. This is an important biological question to be addressed. The authors showed behavior of mice changed during days towards longer immobility duration in the repeated tail suspension exposure, which provides a great way to study "habituation to repeated stress". The claim "Rather, immobility-preferring neurons activated more often" that in Line 294 of the discussion is exciting. But these are nowhere to be found in the results section. The related data and clear interpretation should be provided in the manuscript.

2) It is appreciated that the authors revealed histological verification of lens locations in mPFC of individual mouse. It would be helpful to provide a zoomed in view to show the GCaMP6f-labeled neurons in the layer V of mPFC. In addition, the accuracy of Thy1-GCaMP6f mouse to correctly label pyramidal neurons in mPFC should be demonstrated.

3) In the Fig.1a, the authors showed that mice were exposed to sequences of homecage and tail suspension on days 1,2,3 and 9 with imaging on days 1,3 and 9. Why the tail suspension was performed on days 2 without imaging. Does the tail suspension on day 2 influence the performances of mice on the following days?

4) The obtained results were rarely analyzed by sex even when both male and female mice were used in the experiments. Sex differences in the coping styles have been reported by numerous literatures. Is there any difference in mPFC coding regime during repeated TS exposure between male and female mice. This is an aspect the authors have to discuss in more detail.

References

[1] R F Thompson, W A Spencer. Habituation: a model phenomenon for the study of neuronal substrates of behavior. Psychol Rev. 1966 Jan;73(1):16-43.

[2] Nicola Grissom, Seema Bhatnagar. Habituation to repeated stress: get used to it. Neurobiol Learn Mem. 2009 Sep;92(2):215-24. 

Reviewer #3: This is an interesting manuscript analyzing the neural population dynamics during a behavior where animals are suspended and then they transition between struggling and immobility behaviors. Recordings are then done in layer 5 of the mPFC using Thy1 mice. Neurons are reportedly tracked over days. Population dynamics are then mapped onto a low dimensional manifold using isomap. Notably, the transitions between the two behavioral states can be detected in population quite stably over time. Strikingly, the transition manifold can be realigned across animals into some invariant pattering. However, the manuscript as written raises a number of questions and concerns, especially the lack of explanation about temporally conserved prefrontal activity patterns based on biological individual unit level. Overall, the results are interesting and should be of great interest to multiple fields.

1. Movement speed is quite fast relative to calcium transients, would be more persuasive to show examples and more detailed analysis of temporal alignment between calcium signals and motion speed changes. 

2. Could the author explain why they only focus on imaging the deep layer 5 neurons?

3. The rationale for the behavioral testing and imaging timeline (Day1, 3 and 9) is not explained, especially the author compared the TS paradigm with the contextual fear conditioning paradigm (D1, D2 for short-term while one/two weeks later for long-term) in the discussion session. Why did the author perform the TS-task from Day1 to Day3 but not record on Day2? 

4. The TS selectivity seems to be based solely on the average amplitudes of calcium signals rather than detection or counts of calcium transients based on individual neuron level. Couldn't this just detect variation due to the noise caused by movement artifacts or mis-detection of signal via CaImAn? Could the author show neural activity difference of example neurons during baseline or TS sessions? If possible, could the author also count or calculate the rate of calcium transient/event during both sessions to show the selectivity?

5. While the overall stability of population dynamics is interesting, it is not clear if signal neurons are stable across trials and days? Can the authors detect single neuron responses/transients and show if single neurons maintain responses during and across days?

6. Alignment of trials is usually important for estimating low dim manifolds? How was this done?

7. Registration of cells across time with non-sparse recording is challenging. Examples of tracked population templates and single unit activity patterns across sessions with spatial-temporal information would help.

8. For the speed calculation and related decoding, since the behavioral speed was averaged between all body parts. Are there big differences between speed of different body parts? If so, will this difference also affect the decoding efficiency?

9. Figure 1f: Linear mixed effects modelling is required for this. This will identify contributions and effects per animal. 

10. Figure 2, 3 and 4: All decoding is based on analysis of neural population activity, could the author perform single unit analysis or offer examples of individual unit activity, such as plot of co-firing patterns across neurons in response to changes in behavioral context/coping styles/behavioral speed, or stable tracking of sample neurons activity across days. 

Minor Issue:

11. ExFig 1 & lens implantation: Better anatomical characterization of the brain regions and the lens implantation position are suggested. According to the implantation coordinates and lesion figure offered, with the reference from Allen Institute Adult Mouse Brain Atlas, it seems that the grin lens was located slightly shallower than author's expectation, which seems to be between the edge of M2/premotor cortex and Cg1/cingulate cortex rather than PrL/prelimbic cortex. Besides, for the diameter of the grin lens, it is possible that the imaging neurons also contains part of neurons from M2, which might also contain or encode motion/movement creatures as well as innate escaping/surviving behaviors.

12. Fig 1g, h: It is difficult to see the individual data points and changes across days/comparison groups since the plots are so small.

---

## [Decision Letter · Decision Letter 2]

6 Dec 2023

Dear Dr Sauer,

Thank you for your patience while we considered your revised manuscript "Time-invariant prefrontal activity patterns during repeated exposure to intense threat" for publication as a Short Reports at PLOS Biology. This revised version of your manuscript has been evaluated by the PLOS Biology editors, the Academic Editor and one the original reviewers.

Based on the reviews and on our Academic Editor's assessment of your revision, we are likely to accept this manuscript for publication, provided you satisfactorily address the remaining data and other policy-related requests.

* We would like to suggest a different title to improve accessibility: "Neuronal tuning to threat exposure remains stable in the mouse prefrontal cortex over multiple days"

* DATA POLICY:

Regardless of the method selected, please ensure that you provide the individual numerical values that underlie the summary data displayed in the following figure panels as they are essential for readers to assess your analysis and to reproduce it: 1C, 1D, 1E, 1G, 1H, 2A, 2B, 2D, 2F, 3B, 3C, 3F, 4C and similar figures in the supplementary information. 

* CODE POLICY

Per journal policy, as the code that you have generated is important to support the conclusions of your manuscript, we require that you make it available without restrictions upon publication. Please ensure that the code is sufficiently well documented and reusable, and that your Data Statement in the Editorial Manager submission system accurately describes where your code can be found. If you decide to use github for code and/or data deposition, please assign a DOI so that the repository is citable and versioned for your paper. Zenodo is one of the available tools for this.

We expect to receive your revised manuscript within two weeks. 

*Published Peer Review History*

*Press*

Sincerely,

Christian

Christian Schnell, PhD

Senior Editor,

cschnell@plos.org,

PLOS Biology

Reviewer remarks:

Reviewer #3: The authors have done a good job with revisions. Congratulations on an intriguing paper.

---

## [Editor Report · Decision Letter 3]

19 Dec 2023

Dear Dr Sauer,

Thank you for the submission of your revised Short Reports "Neuronal tuning to threat exposure remains stable in the mouse prefrontal cortex over multiple days" for publication in PLOS Biology. On behalf of my colleagues and the Academic Editor, Jozsef Csicsvari, I am pleased to say that we can in principle accept your manuscript for publication, provided you address any remaining formatting and reporting issues. These will be detailed in an email you should receive within 2-3 business days from our colleagues in the journal operations team; no action is required from you until then. Please note that we will not be able to formally accept your manuscript and schedule it for publication until you have completed any requested changes. 

One of the requested changes will be to link to the source data in the figure legends. For example, by adding 'The data underlying this figure can be found at https://doi.org/10.5281/zenodo.10378756.' to the corresponding figure legends.

PRESS

Sincerely, 

Christian

Christian Schnell, PhD, PhD

Senior Editor

PLOS Biology

cschnell@plos.org